# Using Self-Assembling ADDomer Platform to Display B and T Epitopes of Type O Foot-and-Mouth Disease Virus

**DOI:** 10.3390/v14081810

**Published:** 2022-08-18

**Authors:** Chaowei Luo, Quanhui Yan, Juncong Huang, Jiameng Liu, Yuwan Li, Keke Wu, Bingke Li, Mingqiu Zhao, Shuangqi Fan, Hongxing Ding, Jinding Chen

**Affiliations:** 1College of Veterinary Medicine, South China Agricultural University, No. 483 Wushan Road, Tianhe District, Guangzhou 510642, China; 2Key Laboratory of Animal Vaccine Development, Ministry of Agriculture and Rural Affairs, Guangzhou 510642, China; 3Key Laboratory of Zoonosis Prevention and Control of Guangdong Province, Guangzhou 510642, China

**Keywords:** ADDomer, virus-like particles, foot-and-mouth disease, dominant antigenic epitopes, immunogenicity

## Abstract

Foot-and-mouth disease virus (FMDV) is a highly contagious and devastating virus that infects cloven-hoofed livestock and various wildlife species. Vaccination is the best measure to prevent FMD. ADDomer, as a kind of non-infectious adenovirus-inspired nanoparticle, has the advantage of high thermal stability. In this study, two dominant B-cell antigen epitopes (residues 129~160 and 200~213) and a dominant T-cell antigen epitope (residues 16~44) of type O FMDV were inserted into the ADDomer variable loop (VL) and arginine–glycine–aspartic acid (RGD) loop. The 3D structure of the recombinant protein (ADDomer-RBT) was simulated by homology modeling. First, the recombinant proteins were expressed by the baculovirus expression system and detected by western blot and Q Exactive mass spectrometry. Then the formation of VLPs was observed under a transmission electron micrograph (TEM). Finally, we evaluated the immunogenicity of chimeric VLPs with a murine model. Bioinformatic software analysis preliminarily corroborated that the chosen epitopes were successfully exposed on the surface of ADDomer VLPs. The TEM assay demonstrated the structural integrity of the VLPs. After immunizing, it was found that FMDV-specific antibodies can be produced in mice to induce humoral and cellular immune responses. To sum up, the ADDomer platform can be used as an effective antigen carrier to deliver antigen epitopes. This study presents one of the candidate vaccines to prevent and control FMDV.

## 1. Introduction

Foot-and-mouth disease (FMD), caused by the FMD virus (FMDV), is a highly contagious and economically devastating disease in cloven-hoofed animals [1,2]. FMDV is an RNA virus that belongs to the genus Aphthovirus of the family Picornaviridae [3,4]. The complete viral capsid protein consists of four structural proteins (VP1, VP2, VP3, and VP4) [5]. Studies have shown that FMDV with multiple epitopes can induce strong immune responses [6]. It has been confirmed that T-cell epitopes (residues 16~44) and B-cell epitopes (residues 129~160 and 200~213) on the VP1 protein can be recognized by the organism and can induce an excellent immune response [7,8]. The outbreak of FMD in the past 15 years has shown that serotype O FMDV is the dominant virus strain [9]. Currently, the primary measure of preventing and controlling FMDV is vaccination. However, the traditional inactivated vaccine has disadvantages such as a short immunization cycle and poor thermal stability [10]. Synthetic peptide vaccines and recombinant protein subunit vaccines require high doses of multiple immunizations to induce T-cell responses in the body [11]. Therefore, it is necessary to develop a safe and effective new genetically engineered vaccine for the prevention, control, and eventual elimination of FMD in the future.

Virus-like particles (VLPs) are special bioparticles free of viral nucleic acid [12]. VLPs can mimic natural viral infection to induce robust and protective immune responses [13]. Due to their nano-sized diameters, VLPs can easily enter lymph nodes through lymphatic vessels, directly stimulating B cells to exert humoral immunity and trigger cytotoxic T-cell responses by presenting a major histocompatibility complex (MHC) [14,15]. Chimeric virus-like particles are modified VLPs, which can act as transfer vehicles to carry foreign proteins or antigen epitopes.

In recent years, the French SNRS research team has developed a nanostructured material named ADDomer that can self-assemble to form VLPs. ADDomer is a protein framework constructed by comparing the conserved regions of different adenovirus type 3 amino acid sequences and by modeling protein structures [16]. ADDomer can display various antigenic polypeptides and spontaneously form stable VLPs by the MultiBac system. It has the advantages of a similar virus size and immunogenicity and does not carry genetic material. Therefore, it has been selected as the next-generation vaccine representing the baculovirus expression system. In addition, a type of vaccine expressing the Chikungunya virus E2 protein by ADDomer VLPs proved to be immunogenic in the pre-clinical trial [16]. In the case of the SARS-CoV-2 pandemic, the Pascal Fender team, as the designer of ADDomer, effectively modified the ADDomer vaccine platform with the glycosylation receptor-binding domain (RBD) of SARS-CoV-2. After immunization, the organism elicited a robust neutralizing antibody titer, indicating that ADDomer will become a new tool for the research and development of SARS-CoV-2 VLPs vaccines [17].

This study focused on designing ADDomer as a foreign antigen carrier platform and simulating the 3D structure of VLPs using biological software. The T-cell-dominant epitope (residues 16–44) and B-cell-dominant epitopes (residues 129–160 and 200–213) in serotype O FMDV structural protein VP1 were embedded into the VL and RGD regions of ADDomer without affecting the assembly of ADDomer. The MultiBac baculovirus expression system expressed the recombinant proteins. After preparing the VLPs, we evaluated humoral and cellular immune responses in mice. To our knowledge, the results of this study verified for the first time the designs of the ADDomer platform that embed the antigenic epitopes of FMDV.

## 2. Materials and Methods

### 2.1. Amino Acid Sequence Alignment, 3D Structure Displaying, and Simulation

The 3D structures of ADDomer and ADDomer-RBT were simulated with the protein homology modeling using adenovirus type 3 as a template (PDB accession no.: 4AQQ) (accessed on 28 January 2021). Modeling software (version 10.1; http://www.salilab.org/modeller/) (accessed on 1 February 2021) and a web server of protein homology modeling (http://swissmodel.expasy.org/) (accessed on 1 February 2021) were exploited for protein simulation, whereas PyMoL (version 4.6; http://www.pymol.org/) (accessed on 1 July 2022) was used to display 3D protein structures and surfaces [18], according to the tutorials in PyMOL Wiki (https://pymolwiki.org/index.php/Main_Page) (accessed on 1 July 2022).

### 2.2. Construction of Recombinant Baculoviruses

DH10Multibac competent cells were preserved by the Laboratory of Veterinary Microbiology and Immunology, South China Agricultural University. The ADDomer sequence was provided with patent No.: US201716088905. The T-cell-dominant epitopes and two B-cell-dominant epitopes of FMDV VP1 were derived from the Southeast Asian epidemic strain O/BY/CHA/2010 (GenBank accession number: JN998085.1) isolated in China. To obtain different tandem forms of FMDV VP1, dominant T- and B-cell epitopes were inserted into the ADDomer VL ring, RGD1 ring and RGD2 ring, to obtain the recombinant protein ADDomer-RBT. After the codon optimization of the above nucleic acid sequence for insect cells, artificial gene synthesis was carried out by Sangon (Sangon, Shanghai, China). The synthetic lines of ADDomer and ADDomer-RBT were inserted into the *Bam*H I and *Hin*d III sites of the pFBDM vector. The recombinant baculovirus DNA was recombined with baculovirus DNA in *E. coli* DH10MultiBac. The recombinant baculovirus Bacmid containing ADDomer and ADDomer-RBT genes was transfected into Sf9 cells with Cellfectin II ^®^Reagent (Invitrogen, Carlsbad, CA, USA) to obtain recombinant baculovirus Ac-ADDomer and Ac-ADDomer-RBT.

### 2.3. Generation of VLPs

Two independent cultures of Hi5 cells were grown to 2 × 10^6^ cells/mL before being infected with either Ac-ADDomer or Ac-ADDomer-RBT with a multiplicity of infection (MOI) of 1. Four days after infection, the culture medium was collected. The cell supernatants were removed by centrifugation at 1000 rpm for 10 min. Next, we added five times the volume of PBS (PH 7.4), vortexed the medium to mix, and placed it on ice for sonication. The ultrasonic conditions were 5 s pulses and 9 s pauses for a total of 30 min (40% power). Then it was centrifuged at 11,000 rpm at 4 °C for 15 min. The supernatants were filtered through a 0.22 µm membrane. Using Beckman SW-41 rotor (Beckman Instruments, Brea, CA, USA), the VLP in the supernatants of 10 mL was collected after ultracentrifugation of sucrose cushion with 30%, 50%, 80% concentration of sucrose at 30,000 rpm at 4 °C for 3.5 h. The precipitated VLP was resuspended in 100 µL phosphate-buffered saline (PBS).

### 2.4. Identification of the Recombinant Proteins by Western Blotting and QE Mass Spectrometry

The precipitated recombinant protein (ADDomer-RBT) was loaded onto 12.5% sodium dodecyl sulfate-polyacrylamide gel. After electrophoresis, the protein was transferred to a methanol-activated polyvinylidene fluoride (PVDF) membrane and electroporated at 200 mA for 70 min. Then the PVDF membranes were incubated overnight at 4 °C, while the FMDV positive sera (Guangdong Yongshun Biopharmaceutical Co., Ltd., Guangzhou, China) was used as the primary antibody, diluting with the primary antibody dilution buffer (Beyotime, Shanghai, China). Rabbit anti-porcine IgG (Abbkine, Wuhan, China) coupled with horseradish peroxidase (HRP) diluted with phosphate-buffered solution (PBST) was used as a secondary antibody for signal detection. The membrane was developed using ImmobilonTM Western ECL Substrate (Millipore, Burlington, MA, USA).

A different SDS gel was run for visualization by Coomassie staining; the molecular weight of the protein in the electrophoresis fraction was determined by the color pre-staining protein molecular weight standard. The protein bands found by image analysis were cut from the gel and then analyzed with matrix-assisted laser desorption/ionization time-of-flight (MALDI-TOF) mass spectrometry. After the protein was reduced and alkylated, trypsin (mass ratio 1:50) was added and digested at 37 °C for 20 h. The enzymatic hydrolysate was desalted and freeze-dried and then dissolved in 0.1% FA solution. The mass spectrometry was obtained with MALDI-TOF mass spectrometer ReflexIII (Brooke, Lexington, KY, USA). The cationic range of ultraviolet laser (336 nm) was 500 mur 8000 Da. The autolysis peak of trypsin was calibrated with the known method. The Mascot software (optional peptide fingerprinting) (Matrix Science, Boston, MA, USA) was used to identify the protein by matching the quality of the experiment with the mass of protein listed in the NCBI protein and SwissProt/treMBL database. After scanning (EpsonExpression1680 scanner), the 2D electrophoresis map and the optical density of its single fragment (rectangle) were measured. The raw file of the mass spectrometry test was searched using Mascot (version 2.2) (Matrix Science, London, UK) software to search the corresponding database shown in Table 1 to obtain the identified protein results.

### 2.5. Characterization of VLPs Using Transmission Electron Microscopy (TEM)

The concentrated VLPs samples were aliquoted into 10 µL and added to the carbon-coated grids. After standing for one minute, the residual samples were removed with filter paper. The grids were stained with 1% uranyl acetate for 2 min and then washed with dd water to remove excess stains. After that, we dried the grids for 6 h. Finally, the grids were observed under a field emission transmission electron microscope (FEI Company, Hillsboro, OR, USA).

### 2.6. Animal Immunization and Sample Collection

The concentration of the collected virions was adjusted to 50 µg per mL with PBS and emulsified with ISA201VG adjuvant at 1:1. Then the physicochemical properties were tested. The design of the animal experiment is shown in Table 2. Four-week-old BALB/c mice were randomly divided into 6 groups (*n* = 5 per group). Mice in Groups 1 and 2 were subcutaneously injected with 50 µg of ADDomer VLPs mixed with ISA201VG adjuvant or 50 µg of ADDomer-RBT VLPs mixed with ISA201VG adjuvant, respectively. Mice in Groups 3 and 4 were subcutaneously injected with added 20 µg nucleic acid enhancement adjuvant (NAA) with ADDomer or ADDomer-RBT.

Mice in Group 5 were subcutaneously injected with a dosage of 200 µL commercial vaccine (type O commercial inactivated vaccine of porcine FMD). As a control, mice in Group 6 were injected with PBS only. Mice in each group were injected twice at intervals of 2 weeks (0 and 2 weeks). Blood samples were collected at 7, 14, 21, and 28 days post-immunization (dpi), and sera were separated, and the antibody titer was detected by indirect enzyme-linked immunosorbent assay (ELISA). Before vaccination, all mice were negative for antibodies against FMDV. This study was conducted by following ethical guidelines for animal care in use in China (IACUC permit No. HNPR-2009-05003).

### 2.7. Evaluation of FMDV-Specific Antibody by Indirect ELISA

Specific antibodies were detected in serum by indirect ELISA. ELISA plates were coated with porcine FMD O-type VP1 protein as antigen. ELISA was performed using the FMDV ELISA Antibody Test Kit (Fender Biotechnology Co., Ltd.; Shenzhen, China). According to the mouse serum of positive serum (Group 5) and negative serum (Group 6) collected on the 28th day, we determined the optimal dilution concentration of the serum to be tested (1:1000). A total of 100 µL of diluted serum was added to the wells and incubated at 37 °C for 30 min. After washing with PBS-0.05% Tween 20, enzyme-labeled goat anti-mouse IgG was added and incubated at 37 °C for 30 min. The HRP signal was detected with tetramethylbenzidine (TMB) substrate. The reaction was stopped by adding 50 µL of 2 M H_2_SO_4_. OD values were determined at 450 nm.

### 2.8. Detection of T Lymphocyte Level by Flow Cytometry

T lymphocytes were isolated from the spleen of mice: spleen tissues were ground, and cells were resuspended in PBS. The percentages of T lymphocytes in the spleen to CD4+, and CD8+ T cells were tested in a flow cytometer. Direct labeled CD3-APC, CD4-FITC, and CD8-PE antibodies were added. The negative control group, CD3 positive control group, CD4 positive control group, and CD8 positive control group were set up at the same time. After placing for 30 min on ice, 400 µL PBS resuspension cells were added. The data were collected by flow cytometry (Beckman Coulter, Pasadena, CA, USA) and analyzed by CytExpert (version 2.2) (Beckman Coulter, Pasadena, CA, USA).

### 2.9. IL-2, IL-4, and IFN-γ Release Assays

The sera of mice from each of the experimental groups were collected 2 weeks after the last immunization. The secretion levels of cytokines IFN-γ, IL-2, and IL-4 in sera were quantified according to the instructions of the commercial ELISA kit (MeiMian, Yancheng, Jiangsu, China).

### 2.10. Statistical Analysis

The qualitative analysis of protein peptides from the data obtained by mass spectrometry was performed by Mascot. A one-way analysis of variance performed using GraphPad Prism software (GraphPad Prism Version 6, GraphPad Software, La Jolla, San Diego, CA, USA) was used to analyze the statistical difference between the groups. A *p* < 0.05 was considered statistically significant (NS, *p* > 0.05, * *p* < 0.05, ** *p* < 0.01, *** *p* < 0.001, **** *p* < 0.0001).

## 3. Results

### 3.1. Three-Dimensional Structural Homology Modeling of Recombinant Proteins

According to the ADDomer crystal structure reported in 2019 and the amino acid sequence of ADDomer provided by patent number: US201716088905, 3D structural homology modeling of ADDomer (PDB accession: 6HCR, added on 28 August 2019) was performed using modeller software (version 10.1; http://www.salilab.org/modeller/) (accessed on 1 February 2021) and then displayed using PyMoL (version 4.6; http://www.pymol.org/) (accessed on 1 July 2022). 

ADDomer subunits appear as pentameric proteins (Figure 1a), and ADDomer VLPs appear as protein scaffolds composed of twelve subunits (Figure 1b). Moreover, dominant antigen epitopes of FMDV inserted in the VL loop and RGD loop of ADDomer can be self-assembled as a subunit of ADDomer and displayed as a pentameric protein (Figure 1c), and the inserted epitopes were displayed on the surface of the subunit. At the same time, the inserted epitopes can be displayed on the surface of the ADDomer protein scaffold (Figure 1d).

### 3.2. Expression and Characterization of Recombinant Proteins

The recombinant genes ADDomer and ADDomer-RBT were inserted into transfer vector pFBDM (Figure 2a). Then, the MultiBac expression system was used to acquire bacmid DNA, which was then transfected in Sf9 cells to generate a virus stock. No specific bands were detected in the cells infected with Ac-ADDomer (Figure 2b). There were particle bands in the cells infected with Ac-ADDomer-RBT around 70 kDa, and no specific bands were detected in the cells infected with wild baculovirus. The results showed that FMDV epitopes were successfully embedded in ADDomer and had good reactivity. A series of fragment ions (b ions and y ions produced by amide bond cleavage) were produced by biological mass spectrometry through collision-induced dissociation or post-source decay. The amino acid sequence information of the polypeptide molecules to be determined was obtained by analyzing these fragment ions. According to the results of QE mass spectrometry, the peptide information of each segment was obtained and compared with the amino acid of the target protein. If a peptide had more than 20 points, this indicated that the peptide was successfully identified. The peptide with the highest score was selected to draw the secondary peak map. Amino acid residues 244–275 ADDomer and 72–80 of ADDomer-RBT mass were identified by spectrometry. Combined with the original secondary map results, the results showed that the two proteins were successfully identified (Appendix A). The recombinant proteins were centrifuged in an ultracentrifuge at 30%, 50%, and 80% sucrose solution. After centrifugation at 30,000 r/min at 4 °C for 3.5 h, protein concentration measurements were performed on protein loops of different density gradients. The layer with the highest protein concentration (30~50%) was taken for SDS-PAGE analysis (Figure 2c).

### 3.3. Transmission Electron Microscopy Analysis

Recombinant proteins were observed and analyzed using a transmission electron microscope (TEM). ADDomer could form VLPs with a diameter of about 17 nm (Figure 3a). The shape of the VLP was composed of polygonal pentagons with bumps on the edges, which was similar to that reported in the literature. ADDomer-RBT could form VLPs with a diameter between 30~35 nm (Figure 3b). The shape was more pronounced than ADDomer, and prominent protuberances were shown on the surface of ADDomer-RBT. The results above showed that the recombinant proteins could be successfully self-assembled to form VLPs in vitro.

### 3.4. FMDV-Specific Humoral Immune Responses

The FMDV-specific antibody responses could be detected in all sera of mice immunized with ADDomer-RBT, ADD-RBT + NAA, or an inactivated vaccine (Figure 4). The antibody level increased rapidly and peaked at 28 dpi. Compared with mice inoculated with inactivated vaccines, the difference between the groups was significant at 28 dpi (*p* < 0.05). It could be found that although the immunogenicity of chimeric VLPs with porcine type O FMDV epitopes was not as good as that of commercial inactivated vaccines, it could induce a humoral immune response. The immune efficacy was only inferior to that of the commercial inactivated vaccine. Compared to NAA or not, adding nucleic acid adjuvant NAA can not generate a more specific antibody. Thus, the results above showed that chimeric VLPs with FMDV epitopes could effectively induce the production of FMDV-specific antibodies and had good immunogenicity in mice.

### 3.5. T Lymphocyte Level Detected by Flow Cytometry

VLPs as a vaccine can be presented to CD4+ T cells and CD8+ T cells in the form of “cross-presentation” through MHC-I molecules, thereby triggering specific cellular immunity in the body to produce extensive protection against the virus. In this study, to explore whether ADDomer as a novel chimeric VLP framework has the function of presenting to CD4 and CD8 cells, the percentages of CD4 and CD8 in the spleen of mice on day 28 after the first immunization were analyzed by flow cytometry. At 28 dpi, the percentage of CD4+ T lymphocytes of mice in the experimental groups was higher than that in the PBS group (Figure 5). The level of CD8+ T lymphocytes of the experimental groups was not significantly higher than in the PBS group, indicating that ADDomer VLPs tended to positively stimulate T helper cells after immunization.

### 3.6. The Levels of Cytokines in Sera from Mice Detected by ELISA

To further evaluate the cellular immune response induced by VLPs, the Th1 (IFN-γ and IL-2) and Th2 (IL-4) concentrations were detected by ELISA. The levels of IL-2 and IL-4 cytokines displayed in VLP-vaccinated mice were higher than those in PBS control mice (Figure 6). In Figure 6a, NAA did not affect the secretion of IL-2. The storage of IL-4 in the sera of mice in each group was higher than that in the PBS group (*p* < 0.05). The results indicated that the level of IL-4 in the ADDomer-RBT group was higher than that in the ADDomer group. Mice in the ADDomer-RBT group demonstrated a higher level of IL-4 than those vaccinated with the inactivated vaccine, whereas the addition of NAA enhanced the stimulation of IL-4 in mice (Figure 6b). The ADDomer-RBT group had significantly higher levels of IFN-γ secretion compared to the PBS group. ADDomer and FMDV epitopes in combination stimulated more IFN-γ secretion than ADDomer alone. No significant differences between all VLP-vaccinated groups and the PBS group were observed except for the ADDomer-RBT group (Figure 6c).

## 4. Discussion

Currently, the prevention and control of FMD in China are mainly carried out through inactivated vaccines [19]. However, the regular commercial inactivated vaccines have disadvantages, such as the risk of allergic reactions triggered by allogeneic proteins, short immunization periods, easy immunogenicity to decline with prolonged preservation, and biosafety concerns [20,21]. In this study, we verified the hypothesis that the vaccine containing only FMDV T and B epitopes can also have an excellent immune effect. From the 3D simulation display structure, it was found that the foreign aid antigen epitopes are well displayed on the surface of ADDomer VLPs. Western blotting and QE mass spectrometry identified that the MultiBac expression system successfully formed recombinant bacmid DNA. Transmission electron microscopy showed that the recombinant VLPs were assembled in vitro. After immunizations, the recombinant proteins could induce the immune system to generate specific humoral and cellular immunity in mice. The data indicated that the recombinant VLPs have the potential utility of acting as a vaccine against serotype O FMDV infection.

An antigen epitope is a short amino acid sequence in protein antigen, which is easier to recognize by MHC molecules. Compared with the whole protein antigen, it can directly induce effective immunity [22]. The most studied protein bearing FMDV epitopes with the best effect is the VP1 protein [23,24,25]. Pan et al. inserted the residues 21–40, residues 141–160, and residues 200–213 of FMDV VP1 into the VP2 outer loop of PPV, and constructed a recombinant adenovirus to express PPV:VLPs (FMDV). Foot-and-mouth disease-specific humoral and cellular immunity can be induced after immunization in mice and pigs, and neutralizing antibodies can be produced by inoculation with the foot-and-mouth disease virus after immunization in pigs [25]. Fang et al. constructed three fragments and selected amino acids in type O FMDV VP1: residues 130–140 and residues 141–160 corresponding to B-cell epitopes, residues 16–44 corresponding to a T-cell epitope. By concatenating six peptides in different ways, it was found that the ordering of antigenic epitopes is correlated with the level of the neutralizing antibody produced in vivo [8]. Preliminary studies have shown that the foot-and-mouth disease virus has some antigenic epitopes that play an important role, and has a good immune effect in animal experiments. As one of the new vectors for future vaccine development, ADDomer displays the antigenic epitopes that play an important role in foot-and-mouth disease, which may solve problems such as easy transportation and stability.

At the early stage of our study, two B-cell-dominant epitopes (residues 129–160 and 200–213) and one T-cell-dominant epitope (residues 16–44) of the porcine VP1 protein of FMDV BY/CHA/2010 were selected. Through the baculovirus expression system, recombinant epitope protein (RBT) was expressed, and specific antibodies were produced after immunizing mice and pigs. The disadvantage is that the expression amount of RBT is small, and the economic benefit is not high. This study aimed to produce chimeric VLP-based vaccines using the ADDomer vaccine platform. It is expected to achieve the effect of an improved yield and exert the advantages of the antigenic epitope itself. After simulating using PyMOL, Modeller, and Swiss model bioinformatics tools without compromising the assembly to form VLPs, the selected three FMDV epitopes were inserted into the VL and RGD (RGD1 and RGD2) regions of ADDomer in three ways. The produced chimeric VLPs could display epitopes at the level of software simulation, which was proved by the TEM assay (Figure 1).

Human adenovirus vectors have been widely used in vaccine development and application for several diseases, and human adenovirus serotype 3 (AD3) dodecahedrons can be produced in baculovirus-infected insect cells [16]. They can retain the ability of adenoviruses to penetrate epithelial cells. The baculovirus expression system has many advantages, such as high expression levels and post-translational modifications that allow the expressed heterologous proteins to be correctly folded and biologically active [26]. ADDomer is an adenovirus dodecahedron-derived multimer and can be obtained in high yields by the MultiBac baculovirus expression system. The ADDomer vaccine platform has been applied to develop the chikungunya vaccine and SARS-CoV-2 vaccine [16,27,28]. This kind of vaccine is easy to produce in large quantities on an industrial scale [29]. We used the MultiBac baculovirus expression system to express the recombinant protein (ADDomer-RBT). From Figure 2, ADDomer-RBT inserted into the B-cell antigen epitopes and T-cell antigen epitopes of FMDV could bind to the FMD positive serum. The successful expression of the ADDomer-RBT was confirmed by QE mass spectrometry. After being purified by sucrose gradient centrifugation, VLPs of the ADDomer-RBT with a diameter of about 35 nm were observed under TEM (Figure 3).

ADDomer protein carrying epitopes of FMDV-VP1 could form VLPs in vitro. ADDomer-RBT VLPs can mediate humoral and cellular immunity during immunization. Previous studies point to NAA as immunostimulatory molecules that better improve the protective efficacy of vaccines [30,31]. In this study, a VLP vaccine and NAA adjuvant combination was simultaneously designed in a mouse immunization experiment to check whether the addition of NAA would enhance the immunogenicity of recombinant ADDomer-RBT VLPs. However, the addition of NAA did not significantly change the production of FMDV-specific antibodies. In Figure 4, FMDV-specific antibodies were induced by ADDomer-RBT VLPs, indicating that the inserted FMDV-dominant epitopes played an essential role in exerting immune effects. VLPs as exogenous antigens, which are taken up by obligate APCs and expressed on the surface of the APC membrane in association with MHC class II molecules, are presented to CD4+ T cells [32]. After processing with nucleated cells such as host virus-infected cells or tumor cells, endogenous antigens are shown on their cell membrane surface to CD8+ cells associated with MHC class I expression [33,34]. In Figure 5, 28 days after the first immunization, the increase in CD4+ T cells in the VLP immunized group compared to those in the PBS group was significant (*p* < 0.5). This result demonstrates that VLPs are exogenous antigens that can stimulate the body to produce cellular immune responses.

Furthermore, T cells can differentiate into effector and memory T cells when stimulated by antigens, and effector T cells are divided into Th1 and Th2 type cells, representing T helper type 1 and type 2 cells, respectively. Th1 cells promote lymphocyte proliferation by releasing IL-2 and IFN-γ, while Th2 cells release cytokines such as IL-4 to promote B cell proliferation and differentiation to produce antibodies [35]. The serum cytokine secretion levels of mice 28 days after the first immunization could be detected by ELISA. These data indicated that ADDomer-RBT VLPs act as immunogens to stimulate the production of Th1 or Th2 cytokines.

## 5. Conclusions

To sum up, the insertion of the FMDV epitopes into the VL region and the RGD region of the ADDomer framework does not affect the assembly of VLPs, and mice immunized with chimeric VLPs produce a specific immune response. Unfortunately, no neutralizing antibody test was carried out due to the limitations of the conditions in this study. To investigate the protective effect of chimeric VLPs against FMDV infection, it is necessary to conduct further validation in the future. Notwithstanding its limitation, this study does suggest that the self-assembled nanoparticle scaffold ADDomer has a good development prospect. The ADDomer vaccine platform can be used as a new FMDV VLPs candidate vaccine carrier and a new tool for FMD prevention and control studies.

## 6. Patents

This section is not mandatory but may be added if there are patents resulting from the work reported in this manuscript.

## Figures and Tables

**Figure 1 viruses-14-01810-f001:**
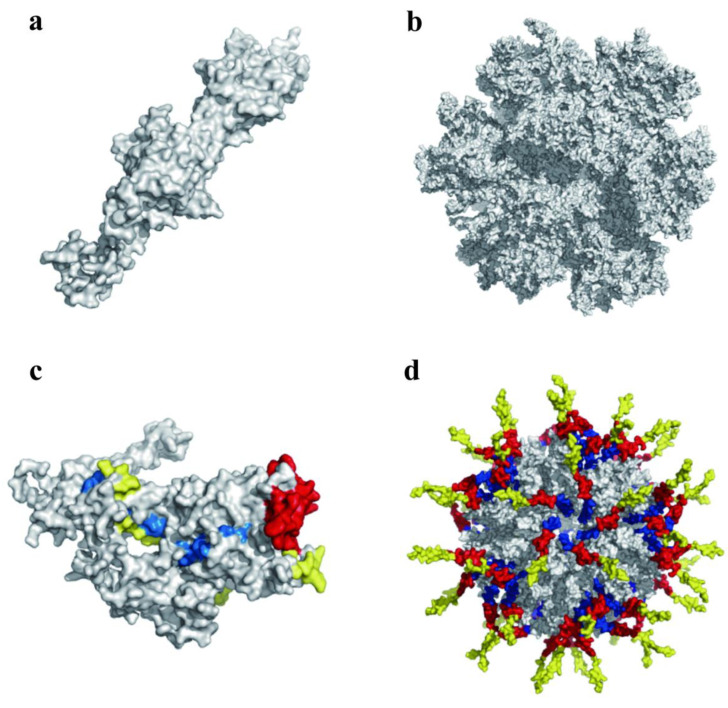
Three-dimensional structure displaying the recombinant ADDomer/ADDomer-RBT subunit and ADDomer/ADDomer-RBT VLP dodecahedron: (**a**) 3D structure of the recombinant ADDomer VLP subunit; (**b**) 3D structure of the recombinant ADDomer VLP dodecahedron. (**c**) Three-dimensional structure of the recombinant ADDomer-RBT VLP subunit; (**d**) 3D structure of the recombinant ADDomer-RBT VLP dodecahedron. Colors yellow, blue, and red represent epitopes inserted in loops VL, RGD1, and RGD2, respectively.

**Figure 2 viruses-14-01810-f002:**
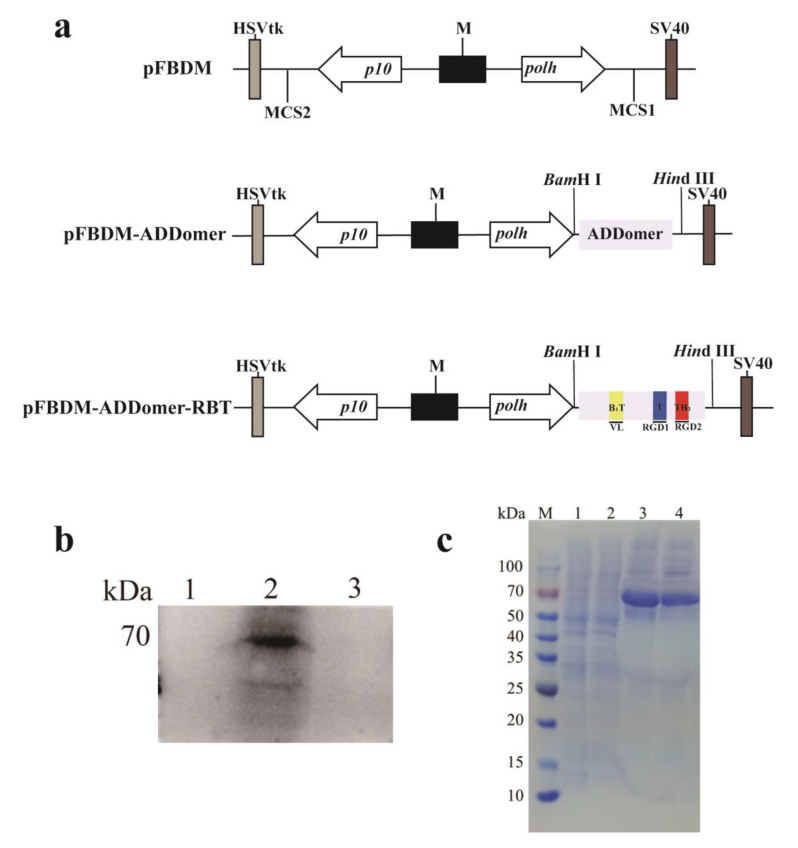
Construction and acquisition of recombinant proteins. (**a**) Construction diagram of pFBDM-ADDomer and pFBDM-ADDomer-RBT recombinant transfer vectors. (**b**) Western blot analysis of recombinant protein expression in Sf9 cells producing recombinant baculovirus. Lane 1: cell lysates of Ac-ADDomer; lane 2: cell lysates of Ac-ADDomer-RBT; channel 3: cell lysates of normal sf9 cells as a negative control. The primary antibody is the FMDV positive sera, and the secondary antibody is the Rabbit anti-porcine IgG-HRP. (**c**) SDS-PAGE analysis of sucrose gradient centrifuged protein loops (30 to 50%, produced by Hi5 cells). Lane 1: ADDomer proteins before sucrose gradient purification. Lane 2: ADDomer-RBT proteins before sucrose gradient purification. Lane 3: after purification of ADDomer proteins. Lane 4: after purification of ADDomer-RBT proteins.

**Figure 3 viruses-14-01810-f003:**
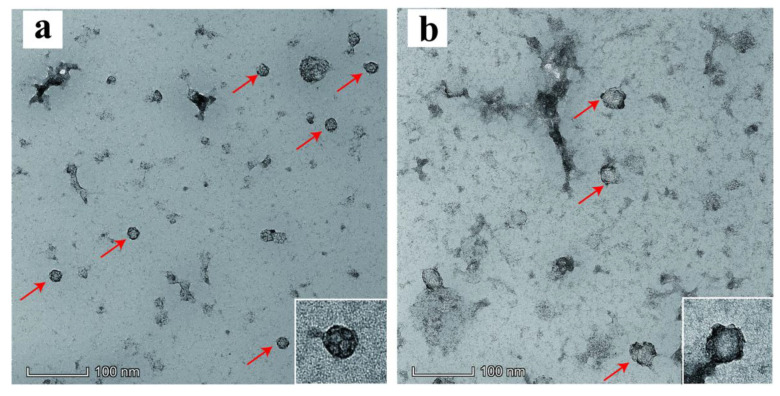
Observation of morphology under a transmission electron microscope (red arrow). (**a**) Recombinant proteins ADDomer; (**b**) recombinant proteins ADDomer-RBT.

**Figure 4 viruses-14-01810-f004:**
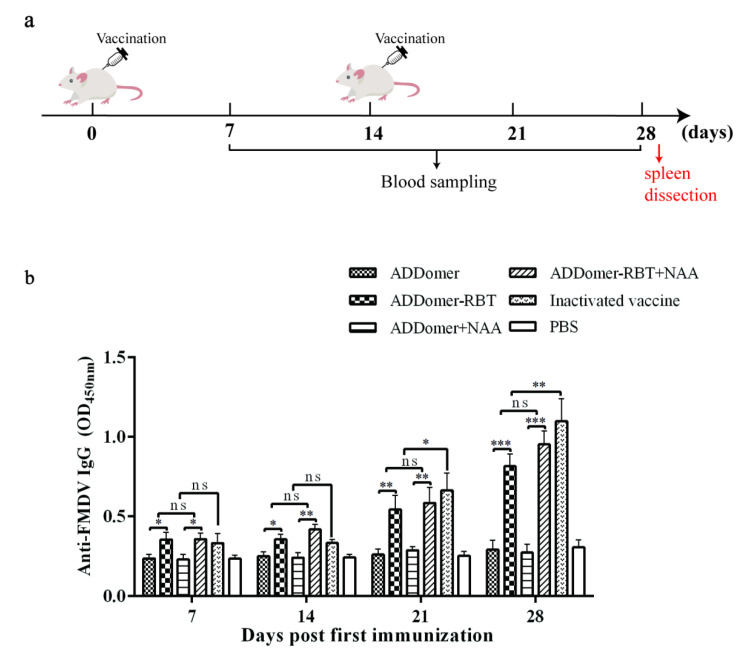
ADDomer-RBT VLPs induced antibody response in mice. (**a**) Immunization procedure of mice. (**b**) Detection of type O FMDV-specific IgG in mice sera (1:1000 dilution) by indirect ELISA using inactivated type O FMDV as coating antigens. Data represent the mean ± SEM. * represents a significant difference between the experimental groups (*p* < 0.05), where * represents *p* < 0.05; ** represents *p* < 0.01; *** represents *p* < 0.001; ns represents no significant difference between the experimental groups (*p* > 0.05).

**Figure 5 viruses-14-01810-f005:**
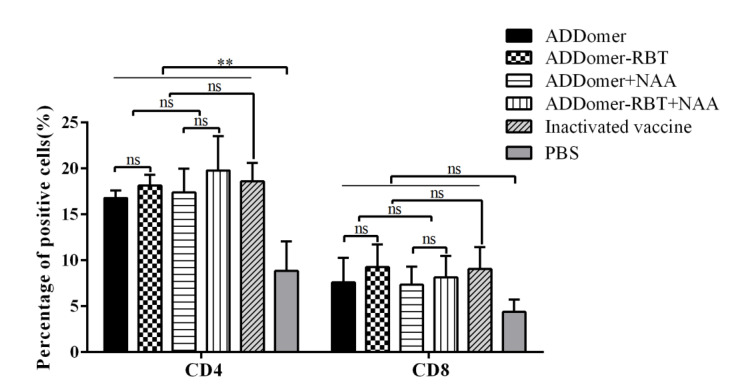
Expression levels of CD4 and CD8 cells were detected by flow cytometry 28 days after the first immunization in mice. Data represent the mean ± SEM. ** represents *p* < 0.01; ns represents no significant difference compared with PBS group (*p* > 0.05).

**Figure 6 viruses-14-01810-f006:**
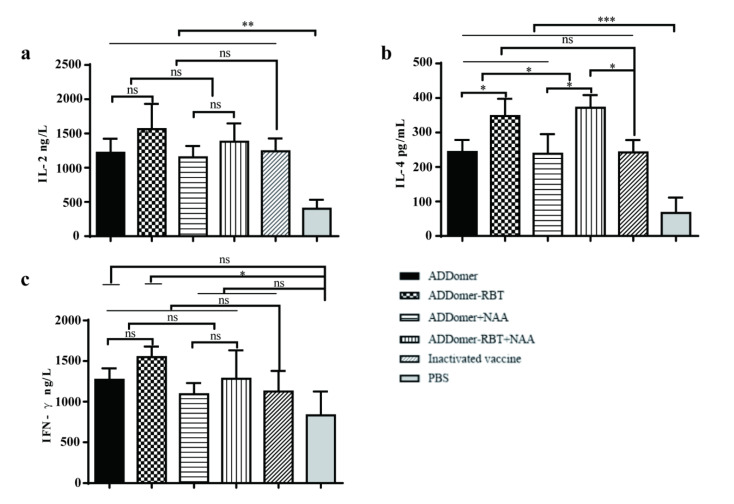
The levels of IL-2, IL-4, and IFN-γ cytokines in sera from mice detected by ELISA. Cytokines in sera were measured in serum samples at 2 wk post last immunization. (**a**) Level of IL-2 in sera; (**b**) level of IL-4 in sera; (**c**) level of IFN-γ in sera. Data represent the mean ± SEM. * represents significant difference from the PBS group (*p* < 0.05), where * represents *p* < 0.05; ** represents *p* < 0.01; *** represents *p* < 0.001; ns represents non-significantly different from PBS group (*p* > 0.05).

**Table 1 viruses-14-01810-t001:** Parameter table of database retrieval.

Enzyme	Trypsin
Database	zjk_1_20201125
Fixed modifications	Carbamidomethyl (C)
Variable modifications	Oxidation (M)
Missed cleavage	2
Peptide mass tolerance	20 ppm
Fragment mass tolerance	0.1 Da
Filter by score ≥ 20	

**Table 2 viruses-14-01810-t002:** Design of animal experiment.

Groups	Number of Mice	Recombinant Proteins	Type and Composition	Immunization	Number of Injection	Immune Dose
1	5	ADDomer	W/O/W	SC	2	200 µL
			ISA 201 VG			
2	5	ADDomer-RBT	W/O/W	SC	2	200 µL
			ISA 201 VG			
3	5	ADDomer	ISA 201 VG + NAA	SC	2	200 µL
4	5	ADDomer-RBT	ISA 201 VG + NAA	SC	2	200 µL
5	5	Inactive vaccine	Commercial vaccine	SC	2	200 µL
6	5	PBS	PBS	SC	2	200 µL

SC: subcutaneous; W/O/W: water-in-oil-in-water emulsion.

## Data Availability

Not applicable.

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
