# Peer review of "Using Self-Assembling ADDomer Platform to Display B and T Epitopes of Type O Foot-and-Mouth Disease Virus"

_viruses, 2022, doi:10.3390/v14081810_

Round 1

Reviewer 1 Report

Please see the attached document

Author Response

Dear editors:

Thank you very much for your valuable advice, it was a huge help for our article. It also gave us a good learning opportunity. Based on your valuables, we have made a lot of revisions to this article. Details have been uploaded to the latest manuscript. The ones with green color are the ones we re-modified. Please see attachment for details. Best wishes!

Author Response

Dear editors:

We apologize for our poor use of English and thank you for your careful reading. At your suggestion, we have revised the manuscript extensively, those highlighted in green are re-edited. Please see attachment for details

Commen 1:Please provide more detail on the origin/preparation of the VP1 ELISA antigen.

Response: Thank you for your comment. The detection of FMDV-specific antibody by indirect ELISA. Use of the Commercial test kits (Fender Biotechnology Co., Ltd; Shenzhen, China; code number:E503104). We refer to this literature method for purchase detection(DOI:10.27152/d.cnki.ghanu.2019.001.001315).

Commen 2: In Figure 1, the images at A and C look very different. Are they viewed from the same angle? And if not, why not?

Response: Thank you for your comment. Images A and C represent the original subunit structure of ADDomer and the subunit structure after epitope insertion, respectively. We are using homology modeling, so the simulated structure will be different from the real structure. Therefore, the simulated structure after the amino acid is inserted into the original structure is only used for comparison with the original structure, and is only used as a reference.

Commen 3:How were the doses of antigen for the mice chosen? The amount of conventional antigen seems high?

Response:Thank you for your comment. For the doses of antigen for the mice chosen, we refer to the literature on the use of porcine circular Cap as a vector for chimeric VLPs, and the antigen dose used to immunize mice in this article is 40ug per mouse[1]. Why are so many doses immune? That's because in addition to antigenic epitopes, chimeric VLPs also have chimeric frameworks. Although 50ug of recombinant protein is used in the article, the proportion of antigenic epitopes in chimeric VLPs is about 16.5%.

Commen 4: Some of the choices of references to general aspects of FMD are not the most relevant choices

Response: Thank you for your comment. We apologize for the confusion generated by the previous version of the manuscript and sincerely hope that our logic is now easier to follow with this new version.

References:

  1. Zhang, H.; Qian, P.; Liu, L.; Qian, S.; Chen, H.; Li, X. Virus-like particles of chimeric recombinant porcine circovirus type 2 as antigen vehicle carrying foreign epitopes. Viruses2014, 6, 4839-4855, doi:10.3390/v6124839.

Reviewer 3 Report

This paper is very poorly written.  There is too much unnecessary verbiage, inconsistencies such as using the terms CpG’s and NAA interchangeable. The other reviewer has marked out.  The Scientific english and sentence construction needs a lot of work.  I would strongly suggest that they send this to a professional scientific writer to clean it up.

The authors overstate the antigenic characteristics of their lead ADDomer platformed construct and over sell the potential of this novel platform. Immune response in mice is only a first step in investigating the potential of a vaccine platform for use in livestock.

The paper should be shortened, there is no need for the extensive reviews as it distracts the reader from the main message which is that ADDomer based technology should be considered as a potential platform next generation vaccines.

Overall, the experimental design is fine, the concept is innovative, however, as with many new platforms, the performance when tested in animals fail to compete with the classic livestock medical countermeasures.  The results and the design concept is of interest to the FMDV community and vaccinologists if better presented.  

Author Response

Dear editors:

Thank you very much for your valuable advice and I apologize for my original poor version. According to your suggestion, the latest version has been modified, green is modified.  Please see attachment for details. We sincerely hope that our logic is now easier to follow with this new version.

Round 2

Reviewer 3 Report

Please look at comments and suggested word substitutions.  

Author Response

Dear editor:

Many thanks for your valuable advice. The main problems have been revised in the newly uploaded manuscript, and the blue color is the latest revision. For the problem of pig positive serum in Western Blot, our experimental process will filter. Thank you for pointing it out. Best wishes for you!

This manuscript is a resubmission of an earlier submission. The following is a list of the peer review reports and author responses from that submission.

Round 1

Reviewer 1 Report

The authors used a self-assembling ADDomer technology to prepare a chimeric VLPs to display B and T epitopes of type O FMDV. The formation of VLPs was observed by TEM. The production of "FMDV-specific" antibodies and activation of CD4 and CD8 T cells were detected. Unfortunately, the methods used in this study are not FMDV-specific at all. For humoral immune response, only ELISA was conducted. What FMDV-specific antigen is used is not indicated. A more specific neutralization assay should be conducted to test the vaccination boosted antibodies' titers to neutralize the FMDV-O infection. Likewise, instead of testing the percentage of CD4 and CD8 cells, a Virus-specific or antigen-specific T cell response should be tested. Whether the T cell response is protective or destructive is also an interesting question. Overall, this study provides limited data on the expression of FMDV epitopes, assembly of ADDomer, and activation of FMDV specific humoral and T cell responses. Moreover, almost all data are observational.

Major points:

There are numerous logic errors and grammar errors. For example, in lines 67-68, "After immunization, the team produced a robust neutralizing antibody titer, which became a new idea for researching and developing SARS-CoV-2 VLPs vaccines[17].", "The team produced titer" does not make any sense. What does "which" mean?

The authors should explain very briefly what exactly ADDomer is. An adenovirus-derived multimeric protein-based self-assembling nanoparticle scaffold should be mentioned.

How did the authors choose epitopes? Is it possible to design a non-sense epitope (or epitope of another virus) as a negative (non-specific) control?

Figure 2, are there any epitope-specific antibodies that can be used to detect the expression of the B and T cell epitopes by Western Blot?

Figure 3, An immune electron microscope may be more powerful in proving that the observed particles expressing FMDV specific epitopes.

Figure 4, The Y-axis title in Figure 4 was cut in half; it is not readable. In addition, the description of the experiments in the legend and the method do not provide many details, which makes it confusing to understand the figure data.

Figure 5 is the same as Figure 4. What are the implications of lymphocyte levels of immunized mice with different ADDomer expressing different epitope peptides? A Virus-specific or antigen-specific T cell response should be tested.